# Progestogens for maintenance tocolysis in symptomatic women. A systematic review and meta-analysis

Francesca Ferrari[1], Silvia Minozzi[2], Laura Basile[1], Giuseppe Chiossi[1], Fabio Facchinetti[1]*

**1** Department of Medical and Surgical Science of the Infant and Adult, University of Modena and Reggio Emilia, Modena, Italy, **2** Department of Epidemiology, Lazio Regional Health Service, Rome, Italy

* facchi@unimore.it

## Abstract

### Objective

Prevention of preterm birth (PTB) with progestogens after an episode of threatened preterm labour is still controversial. As different progestogens have distinct molecular structures and biological effects, we conducted a systematic review and pairwise meta-analysis to investigate the individual role played by 17-alpha-hydroxyprogesterone caproate (17-HP), vaginal progesterone (Vaginal P) and oral progesterone (Oral P).

### Methods

The search was performed in MEDLINE, ClinicalTrials.gov and the Cochrane Central Register of Controlled Trials (CENTRAL) up to 31 October 2021. Published RCTs comparing progestogens to placebo or no treatment for maintenance tocolysis were considered. We included women with singleton gestations, excluding quasi-randomized trials, studies on women with preterm premature rupture of membrane, or receiving maintenance tocolysis with other drugs. Primary outcomes were preterm birth (PTB) < 37 weeks' and < 34 weeks'. We assessed risk of bias and evaluated certainty of evidence with the GRADE approach.

### Results

Seventeen RCTs including 2152 women with singleton gestations were included. Twelve studies tested vaginal P, five 17-HP, and only 1 oral P. PTB < 34 weeks' did not differ among women receiving vaginal P (RR 1.21, 95%CI 0.91 to 1.61, 1077 participants, moderate certainty of evidence), or oral P (RR 0.89, 95%CI 0.38 to 2.10, 90 participants, low certainty of evidence) as opposed to placebo. Instead, 17-HP significantly reduced the outcome (RR 0.72, 95% CI 0.54 to 0.95, 450 participants, moderate certainty of evidence). PTB < 37 weeks' did not differ among women receiving vaginal P (RR 0.95, 95%CI 0.72 to 1.26, 8 studies, 1231 participants, moderate certainty of evidence) or 17-HP (RR 0.86, 95% CI 0.60 to 1.21, 450 participants, low certainty of evidence) when compared to placebo/no

**Data Availability Statement:** All relevant data are within the manuscript and its Supporting Information files.

**Funding:** The author(s) received no specific funding for this work.

**Competing interests:** The authors have declared that no competing interests exist.

treatment. Instead, oral P significantly reduced the outcome (RR 0.58, 95% CI 0.36 to 0.93, 90 participants, low certainty of evidence).

## Conclusions

With a moderate certainty of evidence, 17-HP prevents PTB < 34 weeks' gestation among women that remained undelivered after an episode of threatened preterm labour. However, data are insufficient to generate recommendations in clinical practice. In the same women, both 17-HP and vaginal P are ineffective in the prevention of PTB < 37 weeks'.

## Introduction

Spontaneous preterm birth (PTB) is due to delivery at less than 37 completed weeks' gestation, and it remains the leading cause of neonatal mortality and morbidity in western countries [1].

Most scientific Societies recommend tocolysis to treat spontaneous preterm labour [2]. Acute tocolysis over the first 48 hours is an established practice to delay delivery until antenatal corticosteroids are administered and/or pregnant women are transferred to a tertiary care center. The efficacy of tocolysis beyond the 48 hours (also known as maintenance or secondary tocolysis) is still debated [3]. Indeed, many of the agents tested (magnesium, beta-agonists, etc..) provided no efficacy [4].

Progestogens were found to be useful to prevent PTB in 2 at risk categories: a) women with a prior history of PTB and/or late pregnancy loss [5], and b) women with short cervix at mid gestation [6]. However, most of the PTBs occur among women presenting with signs and symptoms of preterm labour. Although 75% of them remain undelivered after hospital admission, their risk of PTB remains high, as up to 30% delivers prior to 37 weeks' gestation [7]. Several plausible mechanisms of progesterone supplementation to prevent PTB have been proposed, including reduced gap-junction formation, oxytocin antagonism (leading to uterine relaxation), maintenance of cervical integrity and anti-inflammatory effects [8].

Progestogens for maintenance tocolysis were shown to effectively delay delivery according to small low quality randomized controlled trials, but not to larger high-quality RCTs [9]. Furthermore, heterogeneous conclusions were drawn by previous meta-analyses, as different drugs were studied while routes of administration were not adequately accounted for [10, 11]. Thus, the objective of this systematic review and pairwise meta-analysis is to evaluate the efficacy of maintenance tocolysis with 17-alpha-hydroxyprogesterone caproate (17-HP), vaginal progesterone (P) or oral P when compared to placebo or no treatment. We decided to investigate the individual role played by each progestogen as they have different molecular structures and their biological effects are mediated by different receptors [12].

## Materials and methods

We performed this systematic review according to the Preferred Reporting Items for Systematic Reviews and Meta-Analyses (PRISMA) statement [13]. The protocol was registered with PROSPERO (registration number: CRD42020219966; http://www.crd.york.ac.uk/PROSPERO/).

### Data sources and searches

Searches were performed in MEDLINE, ClinicalTrials.gov and the Cochrane Central Register of Controlled Trials (CENTRAL) with the use of a combination of keywords and text words

related to "tocolysis", "preterm labour" and "17-alpha-hydroxyprogesterone caproate" "vaginal progesterone", "oral progesterone" from 1966 until 31 October 2021. To locate additional publications, we reviewed bibliographies of identified studies and reviews articles. No restrictions for language or geographic location were applied. The detailed search strategy is reported in the Supplementary Material (S1 Table).

## Study selection

We included randomized controlled trials on singleton gestations that remained undelivered after an episode of preterm labour, and were then randomized to maintenance tocolysis with either 17-HP, oral P or vaginal P as opposed to no treatment or placebo. Threatened preterm labour was homogenously defined, in the different studies, as the simultaneous presence of regular contractions and cervical changes. Exclusion criteria included quasi-randomized trials, maintenance tocolysis in women with preterm premature rupture of membranes, and maintenance tocolysis with other drugs. Two reviewers (F.F., F.F.) independently screened each record retrieved based on titles and abstracts. Potentially relevant studies were acquired in full text and independently assessed for final inclusion by two authors (F.F., F.F.). Any disagreement was discussed with a third author (S.M.).

## Data extraction and analysis

Data abstraction was independently completed by the same 3 investigators. Any disagreement was reviewed and further resolved by discussion. Data abstracted included: number and characteristics of participants (age, parity, cervical length, history of PTB), number of participants and mean gestational age at randomization in the intervention and control groups, frequency of administration and dosages of 17-HP, vaginal P or oral P.

Primary outcomes were: PTB < 34 weeks' and PTB < 37 weeks' gestation. Secondary outcomes were: latency (defined as the time interval from randomization until delivery), birth weight, low birth weight (i.e. ≤2500 gr), perinatal death, admission to the neonatal intensive care unit (NICU), need for oxygen supplementation, neonatal respiratory distress syndrome (RDS), bronchopulmonary dysplasia (BPD), intraventricular hemorrhage of any class (IVH), and necrotizing enterocolitis (NEC).

In case of missing data on the primary outcomes, we contacted study Authors allowing them one month to reply. Two authors (F.F., S.M.) independently assessed risk of bias according to the criteria set out in the Cochrane Handbook for Systematic Reviews of Interventions [14]. The following criteria were considered: sequence generation and allocation concealment (selection bias), blinding of participants and providers (performance bias), blinding of outcome assessors (detection bias), incomplete outcome data (attrition bias), and selective outcome reporting (reporting bias). Disagreement between reviewers was resolved by discussion.

We analyzed dichotomous outcomes by calculating the risk ratio (RR) for each trial with the uncertainty in each result being expressed with a 95% confidence interval (CI). We analyzed continuous outcomes by calculating the mean difference (MD) with 95% CI. In case the number of analyzed participants did not correspond to the number of randomized subjects, we assumed that missing participants were missed at random, i.e. that missing was unrelated to actual values of the missing data. Therefore, we included in the analysis only the available data without any imputation. As we hypothesized a certain degree of heterogeneity among studies due to treatment schedules, criteria of assessing response, risk of bias and other factors which may have affected direction and magnitude of treatment effect, we pooled data using the random effect model for each outcome. Seeking statistical heterogeneity among studies, the Cochrane Q-test was used with a significant threshold of alpha = 0.1 and inconsistency

among studies was quantified by the I-squared statistic [14]; an I square >70% was considered as significant heterogeneity. We planned to perform subgroup analysis for known risk factor of preterm delivery: cervical length (with a cut-off of 25 mm), and previous PTB or late pregnancy loss (with a cut off of 20% of participants with history of PTB). Results are depicted in all figures as conventional meta-analysis forest plots. RevMan 5.4 was used to generate forest plot figures (Review Manager, version 5.4.1 Copenhagen; 2020). We planned to use visual inspection of funnel plots (plots of the effect estimate from each study against the sample size or standard error) to indicate possible publication bias if there were at least 10 studies included in the meta-analysis.

### Grading of evidence

We assessed the overall quality of the evidence for the primary outcomes using the five GRADE domains (study limitations, consistency of effect, imprecision, indirectness, and publication bias) according to the GRADE approach [15]. Based on the above domains, the GRADE system uses the following rating to grade the evidence: High: we are very confident that the true effect lies close to that of the estimate of the effect. Moderate: we are moderately confident in the effect estimate: the true effect is likely to be close to the estimate of the effect, but there is a possibility that it is substantially different. Low: our confidence in the effect estimate is limited: the true effect may be substantially different from the estimate of the effect. Very low: we have very little confidence in the effect estimate: the true effect is likely to be substantially different from the estimate of effect. The existing evidence was summarized in a "Summary of Findings" table (S2 Table) that provides key information about the magnitudes of relative and absolute effects of the interventions, the amount of available evidence and the certainty of available evidence. We used GRADEproGDT software (GRADEpro GDT, McMaster University; 2015).

### Results

After duplicates were removed, a total of 4390 records were found. We acquired 21 articles in full text, as potentially relevant. Four studies were excluded: 2 because they were not RCTs [16, 17], 1 because participants and interventions did not match the inclusion criteria [18], and 1 because we were unable to retrieve the full text [19] (Fig 1 [13]).

We finally included 17 RCTs [9, 20–35] with 2152 participants: 12 studies compared vaginal P to placebo or no treatment [20–24, 26–28, 32–35], 5 tested 17-HP versus placebo or no treatment [25, 29, 31, 32] while 1 compared oral P to placebo30.

The mean participants' age was 27.3 ± 4.4 years. Mean gestational age at randomization was 30.6 weeks' for vaginal P studies and 28.9 weeks' for 17-HP studies. Among studies on vaginal P, 7 employed a dose of 200 mg/day [9, 20, 21, 23, 26, 27, 32], only one used 90 mg/day [35], while 4 used 400 mg/day [22, 28, 33, 34]. Among RCTs on intramuscular 17-HP, 2 used a 250 mg weekly dose [24, 29], 1 employed 500 mg twice weekly [25], while 2 used 341 mg, either every four days [31] or weekly [32]. Finally, the study investigating oral P utilized a daily dose of 200 mg [30].

Four studies were conducted in Iran [26, 28, 33, 34], 2 in Italy [31, 32], 2 in India [22, 30], 1 enrolled women from Switzerland and Argentina [21], while Canada [9], USA [29], Nepal [24], France [25], Turkey [27], Sweden [35], Portugal [20] and Spain [23] contributed with 1 study each. Four studies were multicenter [23, 28, 29, 30].

Six studies were funded by public institutions [26, 28, 29, 31, 33, 34]; the remaining did not report their funding source. Table 1 summarizes the characteristics and results of each trial.

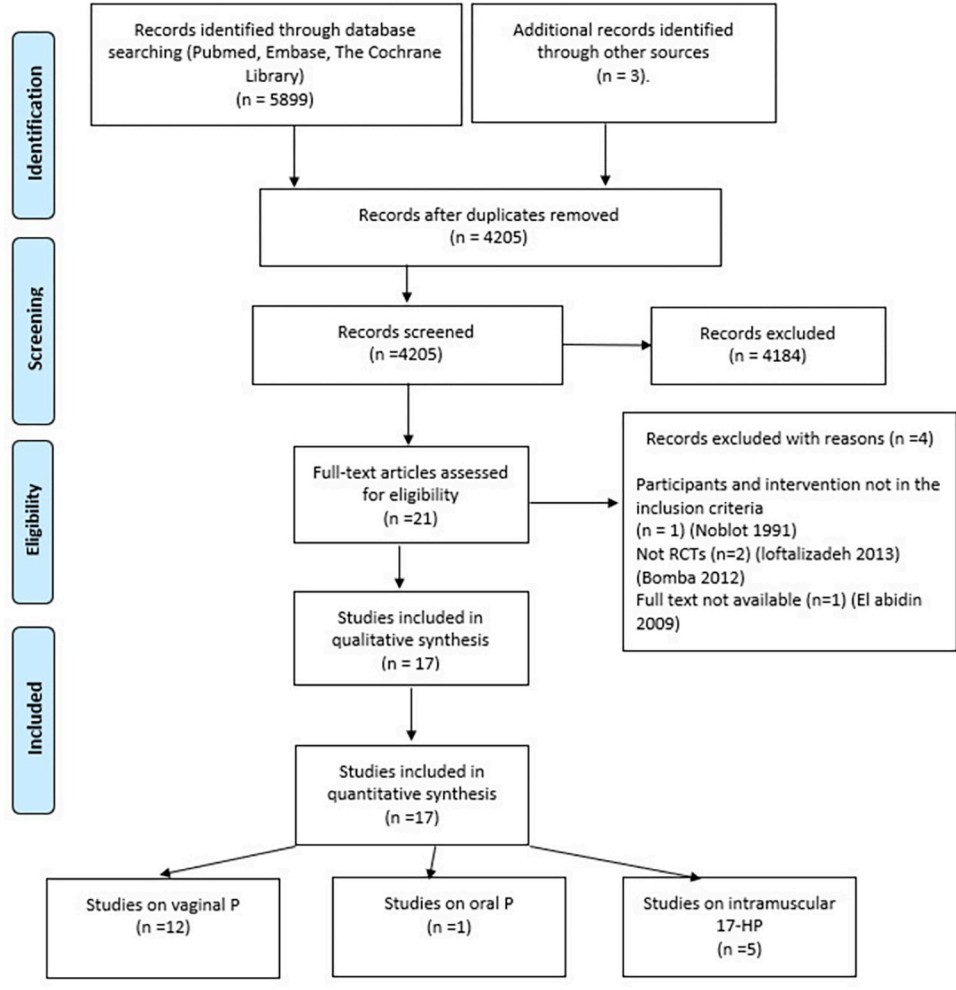

**Fig 1. Flowchart of included studies.**

## Risk of bias of included studies

Seven studies were considered at low risk of selection bias because random sequence generation and allocation concealment were appropriate [9, 24, 26, 28, 29, 31, 33]. Furthermore, random sequence generation was appropriate in 7 [21, 22, 25, 27, 30, 32, 34] studies but information on allocation concealment was not provided. The work of Areia et al was judged at high risk of bias for allocation concealment [20], while Briery et al did not describe how random sequence was generated despite adequate allocation concealment [29]. Eight studies were considered at high risk of performance bias because they were open label [20, 22, 25, 26, 30, 32, 33], while the remaining were judged at low risk as they were placebo controlled, in a double blind design. All the included RCTs were judged at low risk of detection bias because the study outcomes were objective and unlikely to be biased by lack of blinding. Two RCTs were rated high for risk of attrition bias [27, 28]. The study protocol was available only for five studies

**Table 1. Characteristics of included studies.**

| Study | Participants | Additional risk factors | Intervention | Control intervention | Country | Funding | Conflict of interest |
|---|---|---|---|---|---|---|---|
| **Areja 2013** | 52 women with PTL arrested with atosiban Mean age: P → 30.1 ± 4.5 Control → 28.4 ± 5.8 Mean GA at randomization: P → 28.3 ± 2.8 Control → 29.4 ± 2.3 | Parity: NR Previous PTB/late miscarriage: NR Median CL: P → 18.31 (IQR 16–22) Control → 18.5 (IQR 14–23) | Vaginal P (200mg per day) starting immediately after atosiban and continued until delivery. (n = 26) | no treatment (n = 26) | Portugal | NR | none declared |
| **Arikan 2011** | 83 women with PTL arrested with ritodrine Mean age: P → 25.7 ± 4.8 Control → 26.3 ± 5.6 Mean GA at randomization: P → 31.7 ± 2.0 Control → 32.2 ± 2.4 | Parity: NR Previous PTB/late miscarriage: P → 4/43 Control → 3/40 Mean CL: P → 25.6 ± 9.1 Control → 24.9 ± 8.7 | Vaginal P (200 mg per day) starting together with ritodrine, until delivery or 36 weeks and 6 days of gestation. (n = 43) | no treatment (n = 40) | Turkey | NR | none declared |
| **Borna 2008** | 70 women with PTL arrested with magnesium sulphate Mean age: P → 26.1 ± 0.9 Control → 25.5 ± 0.9 Mean GA at randomization: P → 31.1 ± 2.9 Control → 32.4 ± 2.1 | Parity: NR Control → 16/33 Previous PTB/late miscarriage: P → 5/37 Control → 4/33 Mean CL: NR | Vaginal P (400 mg per day) starting within 48 h of arrest of labour and continued until delivery. (n = 37) | no treatment (n = 33) | Iran | NR | none declared |
| **Briery 2014** | 45 women with PTL arrested with intravenous magnesium sulfate, oral calcium channel blockers or antiprostaglandin Mean age: NR Mean GA at randomization: 17-HP → 28.6 ± 2.7 Placebo → 27.3 ± 2.8 | Parity: NR Previous PTB/late miscarriage: NR Mean CL: 17-HP → 26 ± 11 Placebo → 30 ± 11 | 17-HP (250 mg per week) starting after tocolysis was assured by uterine quiescence for 12–24 h and continued until 36 weeks' gestational age. (n = 22) | Placebo (n = 23) | USA | NR | none declared |
| **Choudhary 2014** | 90 women with PTL arrested with nifedipine Mean age: Oral P → 24.1 ± 2.4 Placebo → 23.7 ± 2.9 Mean GA at randomization: Oral P → 31.9 ± 2.1 Placebo → 32.4 ± 1.7 | Parity: Nulliparous Oral P → 16/45 Placebo → 16/45 Previous PTB/late miscarriage: Oral P → 6/45 Placebo → 2/45 Mean CL: NR | Oral P (200 mg per day) starting after 48 hours of acute tocolysis and continued until to 37 weeks or delivery. (n = 45) | Placebo (n = 45) | India | NR | none declared |
| **Facchinetti 2007** | 60 women with PTL arrested with atosiban Mean age: 17-HP → 29.9 ± 3.5 Control → 29.8 ± 2.7 Mean GA at randomization: (days) 17-HP → 208.4 ±22.1 Control → 212.3 ± 18.1 | Parity: Nulliparous 17-HP → 16/30 Control → 17/30 Previous PTB/late miscarriage: 17-HP → 1/30 Control → 2/30 Mean CL: 17-HP → 24.5 ± 8.9 Control → 22.8 ± 9.6 | 17-HP (341 mg every 4 days) after tocolysis and continued until gestational week 36. (n = 30) | no treatment (n = 30) | Italy | NR | none declared |

(*Continued*)

Table 1. (Continued)

| Study | Participants | Additional risk factors | Intervention | Control intervention | Country | Funding | Conflict of interest |
|---|---|---|---|---|---|---|---|
| **Facchinetti 2017** | 254 women with PTL arrested with Atosiban, nifedipine or indomethacin Mean age: 17-HP → 31.8 ± 5.4 P → 32.1 ± 5.7 Control → 31.5 ± 5.9 Mean GA at randomization: 17-HP → 28.4 ± 2.3 P → 28.1 ± 2.8 Control → 28.4 ± 2.5 | Parity: Nulliparous 17-HP → 51/87 P → 45/86 Control → 45/81 Previous PTB/late miscarriage: NR Mean CL: 17-HP → 18.4 ± 4.7 P → 16.7 ± 5.6 Control → 16.6 ± 5.5 | Vaginal P (200 mg per day) or 17-HP (341 mg per week) started after tocolysis and continued until the completion of 36 weeks of gestation. (17-HP n = 87; P n = 86) | no treatment (n = 81) | Italy | funding from both Besins Healthcare and IBSA SA | none declared |
| **Gargari 2012** | 144 women with PTL arrested with magnesium sulfate Mean age: P → 24.2 ± 3.7 Control → 25.4 ± 2.9 Mean GA at randomization: P → 32.2 ± 2.8 Control → 32.7 ± 2.6 | Parity: NR Previous PTB/late miscarriage: NR Mean CL: P → 18 ± 3 Control → 17 ± 4 | Vaginal P (400 mg per day) starting after tocolysis and continued until delivery. (n = 72) | no treatment (n = 72) | Iran | NR | none declared |
| **Hyett 2020** | 84 women with PTL arrested with magnesium sulfate Mean age: P → 27.6 ± 7.6 Placebo → 27 ± 4.8 Mean GA at randomization: P → 29.8 ± 2.3 Placebo → 29.2 ± 1.9 | Parity: Nulliparous P → 36/41 Placebo → 28/33 Previous PTB/late miscarriage: NR Mean CL: P → 21.2 ± 3.7 Placebo → 21.8 ± 3.3 | Vaginal P (400mg per day) starting after tocolysis and continued until until 34 weeks of gestation. (n = 45) | Placebo (n = 35) | Iran | financed and supported by Research ViceChancellor of Shiraz University of Medical Sciences | none declared |
| **Martinez de Tejada 2014** | 385 women with PTL arrested with b-mimetics, oxytocin receptor antagonis or calcium-channel blockers Mean age: P → 27.9 ± 6.5 Placebo → 27.3 ± 5.8 Mean GA at randomization: P → 29.5 ± 2.7 Placebo → 29.5 ± 2.7 | Parity: Nulliparous P → 63/193 Placebo → 73/186 Previous PTB/late miscarriage: P → 48/193 Placebo → 40/186 Mean CL: P → 19.35 ± 8.4 Placebo → 19.6 ± 8.15 | Vaginal P (200 mg per day) starting 48 hours after tocolysis and continued until 36 weeks and 6 days of gestation or until delivery. (n = 193) | Placebo (n = 186) | Switzerland and Argentina | University Hospitals of Geneva, the HUG Clinical Research Center of the Swiss National Foundation, Department of Reproductive Health and Research, World Health Organization, Ministry of Health of Ciudad Auto´noma de Buenos Aires and a personal scholarship for Mrs MC Ocampo | none declared |
| **Mishra 2014** | 100 women with PTL arrested with Isoxsuprine hydrochloride. Mean age: P → 25.1 ± 3.8 Control → 24.1 ± 2.9 Mean GA at randomization: NR | Parity: NR Previous PTB/late miscarriage: P → 22/50 Control → 34/50 Mean CL: NR | Vaginal P (400 mg per day) starting after tocolysis and continued until 36 weeks and 6 days of gestation or delivery. (n = 50) | no treatment (n = 50) | India | NR | none declared |

(*Continued*)

**Table 1.** (Continued)

| Study | Participants | Additional risk factors | Intervention | Control intervention | Country | Funding | Conflict of interest |
|---|---|---|---|---|---|---|---|
| **Palacio 2016** | 265 women with PTL arrested with atosiban Mean age: P → 29.5 ± 5.4 Placebo → 28.5 ± 5.7 Mean GA at randomization: P → 31.7 ± 2.7 Placebo → 31.9 ± 2.3 | Parity: nulliparous P → 67/130 Placebo → 92/135 Previous PTB/late miscarriage: P → 19/130 Placebo → 22/135 Mean CL: P → 17.5 ± 6.5 Placebo → 17.4 ± 6.3 | Vaginal P (200 mg per day) started after tocolysis and continued until delivery. (n = 130) | Placebo (n = 135) | Spain | Montse Palacio was supported by Instituto de Salud Carlos III and Ministerio de Sanidad y Política Social | none declared |
| **Regmi 2012** | 60 women with PTL arrested with Nifedipine. Mean age: 17-HP → 23.2 ± 3.5 Control → 22.8 ± 3.7 Mean GA at randomization: 17-HP → 32.6 ± 1.7 Control → 32.9 ± 1.9 | Parity: Nulliparous 17-HP → 0/29 Control → 0/31 Previous PTB/late miscarriage: 17-HP → 11/29 Control → 20/31 Mean CL: NR | 17-HP (250 mg per week) starting after tocolysis and continued until 37 completed weeks or earlier if they delivered. (n = 29) | no treatment (n = 31) | Nepal | NR | none declared |
| **Rozenberg 2012** | 188 women with PTL arrested with oral nifedipine, intravenous nicardipine, or salbutamol Median age: 17-HP → 29 (IQR 25–34) Control → 29 (IQR 26–33) Median GA at randomization: 17-HP → 28+4 (IQR 26+2–30+2) Control → 27+6 (IQR 26+0–29+3) | Parity: NR Previous PTB/late miscarriage: 17-HP → 15/94 Control → 29/94 Median CL: 17-HP → 17 (IQR 10–20) Control → 15 (IQR 10–20) | 17-HP (500 mg twice a week) starting after tocolysis and continued until 36 weeks or preterm delivery. (n = 94) | no treatment (n = 94) | France | Obtained funding for the study | none declared |
| **Stjernholm 2021** | 58 women with PTL arrested with Atosiban Mean age: P → 31 ± 4 Placebo →29 ± 6 Mean GA at randomization: P → 26 ± 12 Placebo → 26 ± 17 | Parity: Primiparous P → 12/29 Placebo → 16/29 Previous PTB/late miscarriage: NR Mean CL: P → 11 ± 5 Placebo → 12 ± 5 | Vaginal P (90 mg per day) starting after acute tocolysis and continued until 34 + 0 weeks, rupture of the fetal membranes or childbirth (n = 29) | Placebo (n = 29) | Sweden | There was no funding | none declared |
| **Sharami 2010** | 173 women with PTL arrested with magnesium sulfate. Mean age: P → 24.6 ± 5.6 Placebo → 24.1 ± 4.8 Mean GA at randomization: P → 33.5 ± 1.8 Placebo → 33.9 ± 1.5 | Parity: Nulliparous P → 55/86 Placebo → 53/87 Previous PTB/late miscarriage: P → 1/86 Placebo → 3/87 Mean CL: NR | Vaginal P (200 mg per day) starting after 48 hours from tocolysis and continued until 36 weeks gestation. (n = 86) | Placebo (n = 87) | Iran | From Guilan University's vice chancellor | none declared |

(*Continued*)

**Table 1.** (Continued)

| Study | Participants | Additional risk factors | Intervention | Control intervention | Country | Funding | Conflict of interest |
|-------|-------------|-------------------------|--------------|----------------------|---------|---------|----------------------|
| **Wood 2017** | 41 women with PTL arrested with atosiban Mean age: P → 26.5 ± 3.9 Placebo → 29.1 ± 5.9 Mean GA at randomization: P → 29 +0 ± 2+2 Placebo → 28 +4 ± 2+5 | Parity: Nulliparous P → 6/19 Placebo → 10/22 Previous PTB/late miscarriage: P → 7/19 Placebo → 6/22 Mean CL: NR | Vaginal P (200 mg per day) started after acute tocolysis and continued until 35+6 weeks gestation or delivery. (n = 19) | Placebo (n = 22) | Canada | NR | none declared |

PTL, preterm labour; P, vaginal progesterone; Oral P, oral progesterone; 17-HP, 17-alfa hydroxyprogesteron; GA, gestational age; CL, cervical length; PTB, preterm birth; NR, not reported.

[9, 26, 29, 31, 33], where the outcomes reported in the final publication coincided with the ones listed in the original protocol; for taball the remaining studies the protocol was not available, therefore they were considered at unclear risk of selective outcome reporting bias (Fig 2).

## Primary outcomes (S2 Table)

**PTB < 34 weeks.** Preterm birth <34 weeks' gestation did not differ among women receiving vaginal P as opposed to placebo/no treatment (RR 1.21, 95%CI 0.91 to 1.61, 7 studies, 1077 participants, moderate certainty of evidence), or receiving oral P as opposed to placebo (RR 0.89, 95%CI 0.38 to 2.10, 1 study, 90 participants, low certainty of evidence). Instead, 17-HP significantly reduced the outcome when compared to placebo/no treatment (RR 0.72, 95%CI 0.54 to 0.95, 4 studies, 450 participants, moderate certainty of evidence) (Fig 3).

**PTB < 37 weeks.** Preterm birth < 37 weeks' gestation did not differ among women receiving vaginal P as opposed to placebo/no treatment (RR 0.95, 95%CI 0.72 to 1.26, 8 studies, 1231 participants, moderate certainty of evidence), while oral P significantly reduced the outcome when compared to placebo (RR 0.58, 95%CI 0.36 to 0.93, 1 study, 90 participants, low certainty of evidence). Finally, frequency of PTB < 37 weeks did not differ between treatment with 17-HP and placebo/no treatment (RR 0.86, 95%CI 0.60 to 1.21, 4 studies, 450 participants, low certainty of evidence) (Fig 4).

## Secondary outcomes (S1–S21 Figs)

**Latency.** Both 17-HP and vaginal P prolonged time to delivery. The mean difference (MD) in days between randomization to delivery was 12.7 (95%CI 8.32 to 17.08, 9 studies, 806 participants) among women receiving vaginal P when compared to placebo/no treatment (S1 Fig), and 6.64 days (95%CI 1.65 to 11.64, 4 studies, 353 participants) comparing 17-HP to placebo/no treatment (S2 Fig). The study on oral P did not assess this outcome.

**Birth weight.** There were no significant differences in birth weight when vaginal P was compared to placebo/no treatment (MD 130.85 g, 95%CI 12.60 to 274.30, 9 studies, 899 participants) (S3 Fig), while birth weight was higher among those treated with oral P (MD 300.00 g, 95%CI 81.88 to 518.12, 1 study, 90 participants) (S4 Fig), or with 17-HP (MD 134.18 g, 95% CI 21.99 to 246.38, 5 studies, 510 participants) (S5 Fig).

**Low birth weight.** Newborns with low birth weight were significantly reduced in the vaginal P studies compared to placebo/no treatment (RR 0.63; 95%CI 0.45 to 0.88, 7 studies, 678 participants) (S6 Fig) and in the oral P study (RR 0.59; 95%CI 0.37 to 0.94, 1 study, 90

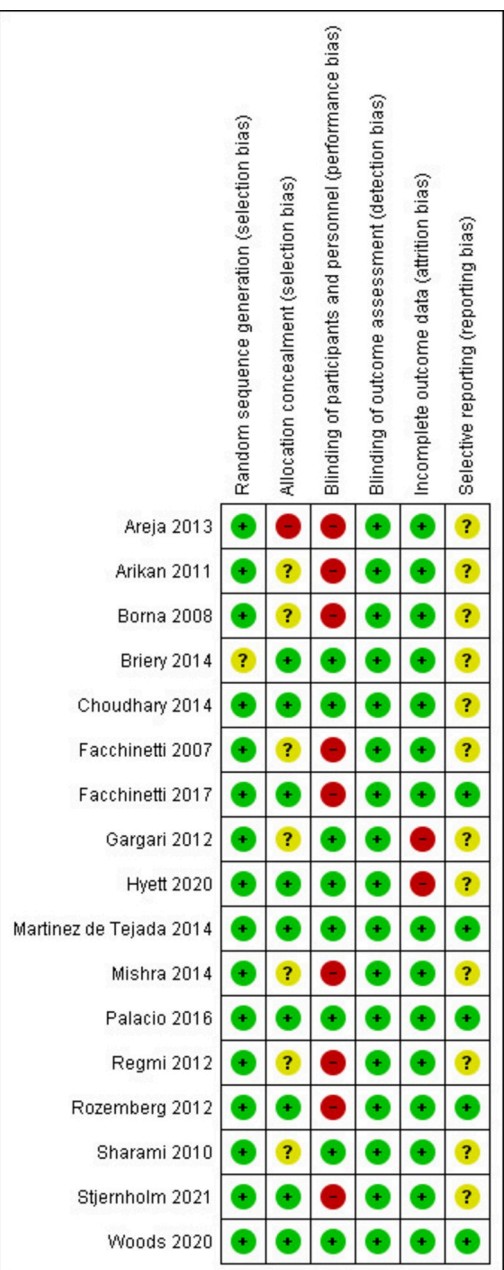

**Fig 2. Risk of bias of included studies.**

participants) (S7 Fig), while no significant differences were found between 17-HP and placebo/no treatment (RR 1.04; 95%CI 0.59 to 1.84, 2 studies, 217 participants) (S8 Fig).

**Perinatal death.** Perinatal deaths were similar among women treated with vaginal P and those receiving placebo/no treatment (RR 0.53, 95%CI 0.25 to 1.1, 7 studies, 952 participants) (S9 Fig). Same finding was observed for oral P (RR 1.00, 95%CI 0.06 to 15.50, 1 study, 90 participants) (S10 Fig), and 17-HP (RR 0.15, 95%CI 0.01 to 2.73, 2 studies, 233 participants) (S11 Fig).

**Admission to neonatal intensive care unit (NICU).** No significant differences were found in NICU admissions in any of the comparisons (vaginal P (S12 Fig): RR 0.8; 95%CI 0.56

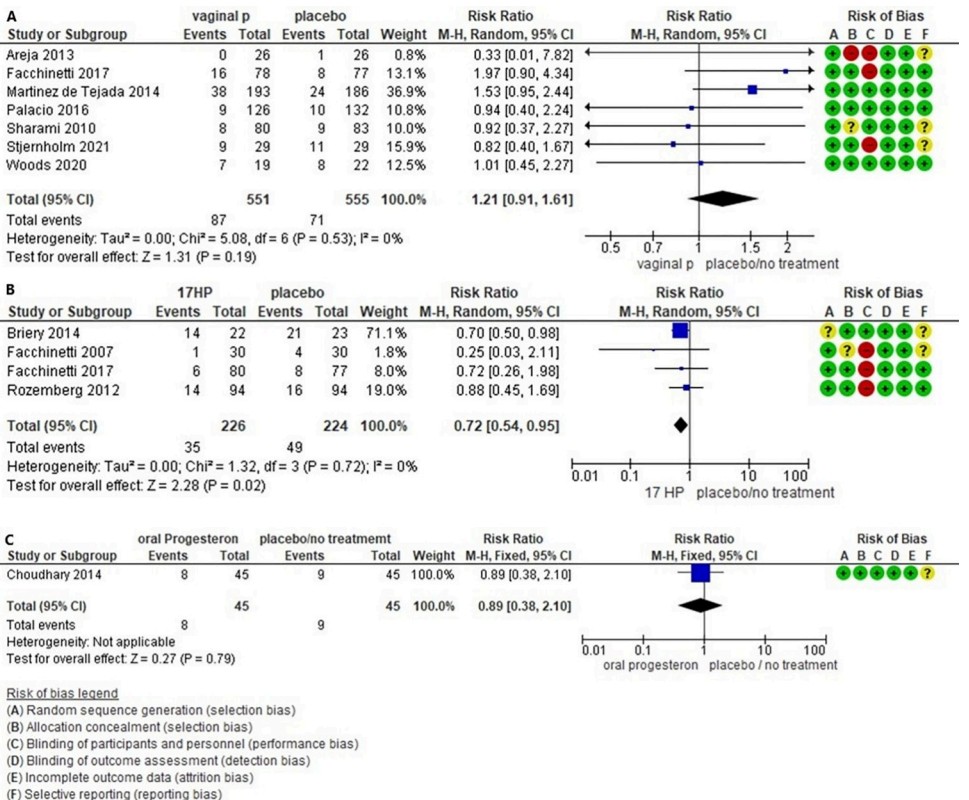

**Fig 3. Forest Plot PTB < 34 weeks.** A. Forest plot comparison (Vaginal P compared with no treatment or placebo) for outcome: preterm birth < 34 weeks' gestation. B. Forest plot comparison (17-HP compared with no treatment or placebo) for outcome: preterm birth < 34 weeks' gestation. C. Forest plot comparison (Oral P compared with no treatment or placebo) for outcome: preterm birth < 34 weeks' gestation. CI, confidence interval; Vaginal P, vaginal progesterone; Oral P, oral progesterone; 17-HP, 17-alfa hydroxyprogesterone.

to 1.15, 9 studies, 1226 participants; 17-HP (S13 Fig): RR 0.93; 95%CI 0.53 to 1.64, 3 studies, 405 participants; oral P (S14 Fig): RR 1.11; 95%CI 0.50 to 2.47, 1 study, 90 participants).

**Neonatal respiratory distress syndrome (RDS).** Fewer RDS diagnoses were made among new-borns whose mothers were in the vaginal P compared to the placebo/no treatment group (RR 0.63, 95%CI 0.42 to 0.95, 7 studies, 541 participants) (S15 Fig), while no significant differences were noted between those receiving oral P (RR 0.86,95%CI 0.31 to 2.35, 1 study, 90 participants) (S16 Fig), or 17-HP (RR 0.98, 95%CI 0.60 to 1.62, 3 studies, 293 participants) (S17 Fig).

**Need of oxygen.** No significant differences were found in the need of oxygen among neonates whose mothers received vaginal P (RR 0.71, 95%CI 0.33 to 1.50, 5 studies, 423 participants) (S18 Fig) or 17-HP (RR 0.24, 95%CI 0.03 to 2.11, 1 study, 157 participants) (S19 Fig), as opposed to placebo/no treatment. The study on oral P did not assess this outcome.

None of the included studies addressed Bronchopulmonary dysplasia (BPD), Intraventricular haemorrhage (IVH) or Necrotizing enterocolitis (NEC).

**Subgroup analysis.** Subgroup analyses for mean cervical length at baseline was not feasible, as 5 studies did not report the measure and all but 2 of the remaining studies reported a mean cervical length less than 25 mm.

Subgroup analysis for history of PTB/late pregnancy loss was possible only when vaginal P was compared to placebo/no treatment (6 studies), while only 1 study investigating 17-HP

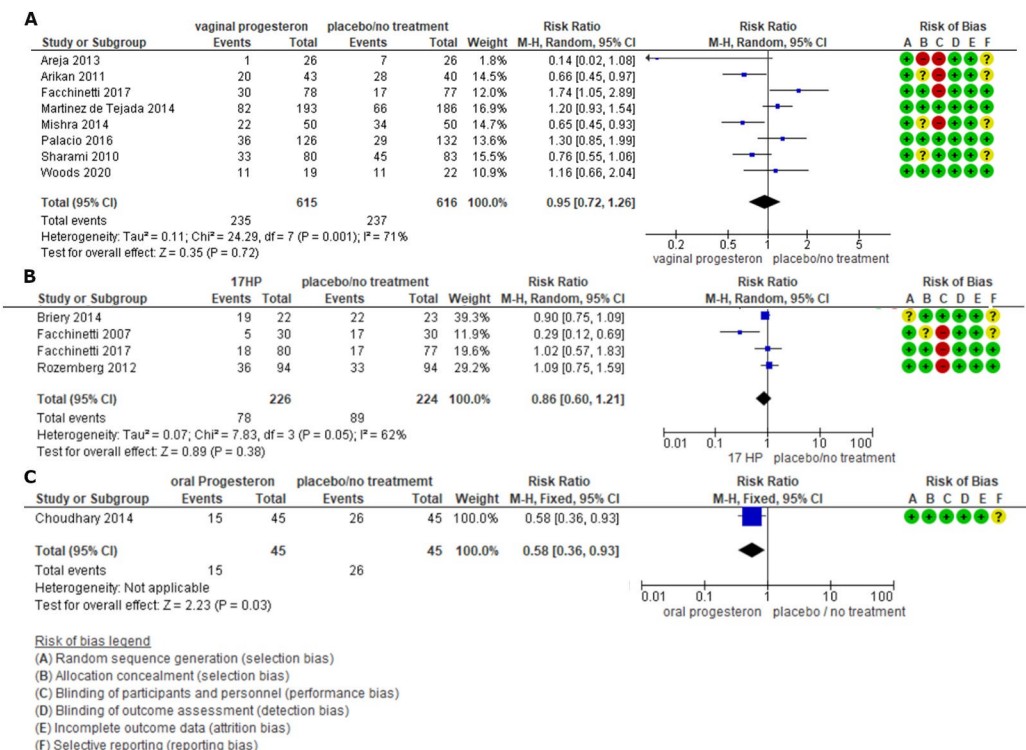

**Fig 4. Forest Plot PTB < 37 weeks'.** A. Forest plot comparison (Vaginal P compared with no treatment or placebo) for outcome: preterm birth < 37 weeks' gestation. B. Forest plot comparison (17-HP compared with no treatment or placebo) for outcome: preterm birth < 37 weeks' gestation. C. Forest plot comparison (Oral P compared with no treatment or placebo) for outcome: preterm birth < 37 weeks' gestation. CI, confidence interval; Vaginal P, vaginal progesterone; Oral P, oral progesterone; 17-HP, 17-alfa hydroxyprogesterone.

reported this information. No significant differences were found in the incidence of PTB < 34- or < 37-weeks' gestation (S20 and S21 Figs respectively) between subgroups of studies including > or ≤ 20% of participants with a history of PTB/late pregnancy loss (Test for subgroup differences: $Chi^2 = 1.04$, df = 1 (P = 0.31), $I^2 = 4.3\%$ and $Chi^2 = 0.15$, df = 1 (P = 0.70), $I^2 = 0\%$, respectively).

## Discussion

This systematic review is the largest up-do-date, including 17 RCTs and 2152 singleton gestations, 410 (23%) women more than previous meta-analysis [9].

The quantitative meta-analysis showed for the first time, with moderate certainty of evidence, that 17-HP given to women remaining undelivered after an episode of threatened preterm labour significantly reduced their risk of PTB <34 weeks' gestation. It should be underlined that the relatively small number of events impacts on the precision of the estimate. We considered PTB <34 weeks' gestation as the primary outcome because clinical management of threatened preterm labour substantially differs after this time point. Acute tocolysis is not recommended in the late preterm period [36] (i.e. delivery between 34 and 37 weeks') [37] while antenatal corticosteroids treatment is still debated [38].

Although our primary outcome had been addressed in previous meta-analyses [9, 39], our conclusions differ due to a larger study population, that now includes data recovered from 2RCTs [31, 32] originally reporting PTB <35 weeks' as primary outcome. This allows almost doubling the number of participants in our actual study.

Despite such positive effect on PTB <34 weeks' gestation, we confirmed that 17-HP did not affect PTB <37 weeks'. The apparent inconsistency may be due to a limitation of primary studies, which do not systematically differentiate between spontaneous and iatrogenic PTB (defined as planned delivery for maternal and/or adverse fetal conditions), the latter being the most common delivery cause in the late preterm period [37].

Based on a large study population (>2000 women) deriving from different countries, our meta-analysis also corroborated the previous suspect that vaginal P is not helpful for PTB prevention after an episode of preterm labour [9, 39, 40]. Nationality represents a pivotal issue, as a secondary analysis of the "4P trial" demonstrated that vaginal P increased the risk of PTB in women from Switzerland while it was ineffective among Argentinians [41], hinting towards potential harmful effects, as already hypothesized by other Authors [9].

Distinct molecular structures, biological effects and sites of actions (genomics and non-genomics) may account for differences in the effectiveness of P and 17-HP on PTB rates [42, 43]. Accordingly, we decided to analyse RCTs using the 2 progestogens separately, and not to combine data on vaginal and oral P, as the 2 routes of administration lead to different metabolites and clearance [44]. Thus, we cannot drew definitive conclusions on oral P since only one, low-quality small trial was included in our review.

Concerning secondary outcomes we confirmed that both 17-HP [11] and vaginal P [10] are associated with prolongation of pregnancy when compared to placebo/no treatment, leading to higher birth-weights, despite unchanged NICU admission rates. The fact the vaginal P treatment is not associated with reduction of PTB < 34 weeks but with significantly increased latency could be related to the gestational age at randomization. Indeed, vaginal P studies randomized women almost two weeks later compared to 17-HP studies. Latency is therefore an outcome conditioned by gestational age at randomization, so it must be interpreted in relation to it. Moreover, the three trials that reported a significant latency did not investigate PTB <34 weeks as an outcome [22, 33, 34]. Finally, it also seems that perinatal mortality and RDS were reduced among women treated with vaginal P, while data on 17-HP showed no difference [11]. The apparent conflicting results between effects of the 2 progestogens on primary outcomes and lack of related effects on morbidity, could be ascribed to the small number of events included in the MA for low birth weight and RDS. The small number of events, together with the small sample size and the relatively small number of included studies strongly affect the precision of the estimate, returning in wide confidence intervals crossing the line of no effect.

The strengths of this systematic review include its comprehensive search strategy and rigorous statistical analysis. Our findings do not solely rely on the higher number of included studies, but also on the systematic search of unpublished data for the two primary outcomes. We performed a subgroup analysis on the impact of additional risk factors for PTB, such as previous PTB/late pregnancy loss, showing for the first time that vaginal P did not affect PTB rates, independently of patients' obstetric history. Furthermore, we found a trend towards higher PTB rates < 34 weeks' gestation when vaginal P was given to populations with previous PTB.

Our work is not without limitations: different potential sources of risk of bias, as well as lack of availability of the majority of study protocols, increased the risk of selective outcomes reporting. This is the reason why we drew definitive conclusions only on the primary outcomes. We could not perform the subgroup analysis on the history of PTB/late pregnancy loss on women treated with 17-HP due to the paucity of studies. Similarly, subgroup analysis for mean cervical length at baseline was not feasible because 5 studies did not report the measure, and all but 2 of the remaining studies reported a mean cervical length < 25 mm. These limitations remind us of the importance of international networks to encourage database exchanges. For instance, the recent Individual Patient Data meta-analysis on the effects of progestogens in

women at risk for positive obstetric history or asymptomatic short cervix had such a large population that its conclusions could impact clinical practice [45].

In conclusion, with a moderate certainty of evidence we found that 17-HP prevents PTB < 34 weeks' gestation among women that remained undelivered after an episode of threatened preterm labour. Vaginal P was ineffective, although it seems to prolong pregnancy. However, there are insufficient data to generate recommendations for the use of progestagens as a maintenance tocolytic in clinical practice. Prior to any change in patients' management, a large multicentre international RCT is urged to detect possible differences in PTB prevention among women treated with vaginal P, intramuscular 17-HP or placebo.

## Supporting information

**S1 Fig. Forest plot comparison (Vaginal P compared with no treatment or placebo) for outcome: Latency.** CI, confidence interval; Vaginal P, vaginal progesterone; SD, Standard deviation.
(TIF)

**S2 Fig. Forest plot comparison (17-HP compared with no treatment or placebo) for outcome: Latency.** CI, confidence interval; 17-HP, 17-alfa hydroxyprogesterone; SD, Standard deviation.
(TIF)

**S3 Fig. Forest plot comparison (Vaginal P compared with no treatment or placebo) for outcome: Birth weight.** CI, confidence interval; Vaginal P, vaginal progesterone; SD, Standard deviation.
(TIF)

**S4 Fig. Forest plot comparison (Oral P compared with no treatment or placebo) for outcome: Birth weight.** CI, confidence interval; Oral P, oral progesterone; SD, Standard deviation.
(TIF)

**S5 Fig. Forest plot comparison (17-HP compared with no treatment or placebo) for outcome: Birth weight.** CI, confidence interval; 17-HP, 17-alfa hydroxyprogesterone; SD, Standard deviation.
(TIF)

**S6 Fig. Forest plot comparison (Vaginal P compared with no treatment or placebo) for outcome: Low birth weight.** CI, confidence interval; Vaginal P, vaginal progesterone; SD, Standard deviation.
(TIF)

**S7 Fig. Forest plot comparison (Oral P compared with no treatment or placebo) for outcome: Low birth weight.** CI, confidence interval; Oral P, oral progesterone; SD, Standard deviation.
(TIF)

**S8 Fig. Forest plot comparison (17-HP compared with no treatment or placebo) for outcome: Low birth weight.** CI, confidence interval; 17-HP, 17-alfa hydroxyprogesterone; SD, Standard deviation.
(TIF)

**S9 Fig. Forest plot comparison (Vaginal P compared with no treatment or placebo) for outcome: Perinatal death.** CI, confidence interval; Vaginal P, vaginal progesterone. (TIF)

**S10 Fig. Forest plot comparison (Oral P compared with no treatment or placebo) for outcome: Perinatal death.** CI, confidence interval; Oral P, oral progesterone. (TIF)

**S11 Fig. Forest plot comparison (17-HP compared with no treatment or placebo) for outcome: Perinatal death.** CI, confidence interval; 17-HP, 17-alfa hydroxyprogesterone. (TIF)

**S12 Fig. Forest plot comparison (Vaginal P compared with no treatment or placebo) for outcome: Admission in neonatal intensive care unit (NICU).** CI, confidence interval; Vaginal P, vaginal progesterone. (TIF)

**S13 Fig. Forest plot comparison (17-HP compared with no treatment or placebo) for outcome: Admission in neonatal intensive care unit (NICU).** CI, confidence interval; 17-HP, 17-alfa hydroxyprogesterone. (TIF)

**S14 Fig. Forest plot comparison (Oral P compared with no treatment or placebo) for outcome: Admission to neonatal Intensive care unit (NICU).** CI, confidence interval; Oral P, oral progesterone. (TIF)

**S15 Fig. Forest plot comparison (Vaginal P compared with no treatment or placebo) for outcome: Neonatal respiratory distress syndrome (RDS).** CI, confidence interval; Vaginal P, vaginal progesterone. (TIF)

**S16 Fig. Forest plot comparison (Oral P compared with no treatment or placebo) for outcome: Respiratory distress syndrome (RDS).** CI, confidence interval; Oral P, oral progesterone. (TIF)

**S17 Fig. Forest plot comparison (17-HP compared with no treatment or placebo) for outcome: Neonatal respiratory distress syndrome (RDS).** CI, confidence interval; 17-HP, 17-alfa hydroxyprogesterone. (TIF)

**S18 Fig. Forest plot comparison (Vaginal P compared with no treatment or placebo) for outcome: Need for oxygen.** CI, confidence interval; Vaginal P, vaginal progesterone. (TIF)

**S19 Fig. Forest plot comparison (17-HP compared with no treatment or placebo) for outcome: Need for oxygen.** CI, confidence interval; 17-HP, 17-alfa hydroxyprogesterone. (TIF)

**S20 Fig. Subgroup analysis for history of PTB (cut off 20% of participants with a history of PTB) PTB < 34 weeks.** CI, confidence interval; Vaginal P, vaginal progesterone; PTD, Preterm delivery; PTB, Preterm birth. (TIF)

**S21 Fig. Subgroup analysis for history of PTB (cut off 20% of participants with a history of PTB) PTB < 37 weeks.** CI, confidence interval; Vaginal P, vaginal progesterone; PTD, Preterm delivery; PTB, Preterm birth.
(TIF)

**S1 Table. Studies selection strategy.** Vaginal P, vaginal progesterone; Oral P, oral progesterone; 17-HP, 17-alfa hydroxyprogesterone; PTB, Preterm Labour.
(DOCX)

**S2 Table. Summary of findings table (SoF).** CI: Confidence interval; RR: Risk ratio; Oral P: Oral progesterone; Vaginal P: Vaginal progesterone; 17-OH: 17 hydroxyprogesterone. High certainty: We are very confident that the true effect lies close to that of the estimate of the effect. Moderate certainty: We are moderately confident in the effect estimate: The true effect is likely to be close to the estimate of the effect, but there is a possibility that it is substantially different. Low certainty: Our confidence in the effect estimate is limited: The true effect may be substantially different from the estimate of the effect. Very low certainty: We have very little confidence in the effect estimate: The true effect is likely to be substantially different from the estimate of effect. Explanations: a. downgraded two levels for imprecision: very few events.
(DOCX)

## Author Contributions

**Conceptualization:** Giuseppe Chiossi, Fabio Facchinetti.

**Data curation:** Francesca Ferrari, Silvia Minozzi, Laura Basile.

**Formal analysis:** Silvia Minozzi, Laura Basile.

**Methodology:** Francesca Ferrari, Silvia Minozzi, Giuseppe Chiossi, Fabio Facchinetti.

**Supervision:** Francesca Ferrari, Fabio Facchinetti.

**Validation:** Fabio Facchinetti.

**Writing – original draft:** Francesca Ferrari, Silvia Minozzi, Laura Basile, Giuseppe Chiossi, Fabio Facchinetti.

**Writing – review & editing:** Francesca Ferrari, Giuseppe Chiossi, Fabio Facchinetti.

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
