## [Decision Letter · Decision Letter 0]

31 May 2022

PONE-D-22-09216PROGESTOGENS FOR MAINTENANCE TOCOLYSIS IN SYMPTOMATIC WOMEN. A SYSTEMATIC REVIEW AND META-ANALYSISPLOS ONE

Dear Dr. Ferrari,

Thank you for submitting your manuscript to PLOS ONE. After careful consideration, we feel that it has merit but does not fully meet PLOS ONE’s publication criteria as it currently stands. Therefore, we invite you to submit a revised version of the manuscript that addresses the points raised during the review process.

We look forward to receiving your revised manuscript.

Kind regards,

Anish Keepanasseril

Academic Editor

PLOS ONE

Journal Requirements:

2. PLOS requires an ORCID iD for the corresponding author in Editorial Manager on papers submitted after December 6th, 2016. Please ensure that you have an ORCID iD and that it is validated in Editorial Manager. To do this, go to ‘Update my Information’ (in the upper left-hand corner of the main menu), and click on the Fetch/Validate link next to the ORCID field. This will take you to the ORCID site and allow you to create a new iD or authenticate a pre-existing iD in Editorial Manager. Please see the following video for instructions on linking an ORCID iD to your Editorial Manager account: https://www.youtube.com/watch?v=_xcclfuvtxQ.

Additional Editor Comments:

The manuscript assess the role of maintenance therapy with two type of progesterone on preventing preterm birth t less than 34 and 37 weeks of gestation.

There are some major concerns raised by the reviewers and its mainly due to the inconsistency of the writing or the language used.

How was the meta- analysis done when there are no direct comparison of these two types of therapy in any of the study included in the analysis?.

Reviewers' comments:

Reviewer's Responses to Questions

**Comments to the Author**

1. Is the manuscript technically sound, and do the data support the conclusions?

Reviewer #1: Partly

Reviewer #2: Yes

Reviewer #3: Yes

2. Has the statistical analysis been performed appropriately and rigorously? 

Reviewer #1: I Don't Know

Reviewer #2: Yes

Reviewer #3: Yes

3. Have the authors made all data underlying the findings in their manuscript fully available?

Reviewer #1: Yes

Reviewer #2: Yes

Reviewer #3: Yes

4. Is the manuscript presented in an intelligible fashion and written in standard English?

Reviewer #1: Yes

Reviewer #2: Yes

Reviewer #3: Yes

5. Review Comments to the Author

Reviewer #1: • First of all how was preterm labor defined in these studies? How many patients were in established preterm labor with cervical changes and how many were only threatened preterm labor in each study?

• Some of the studies have used vaginal P only up to 34 weeks, what is the point of studying delivery at 37 weeks in those studies?

• Subgroup analysis with regards to the initial drug used for arrest of preterm labor would be desirable

• “The apparent inconsistency may be due to a limitation of primary studies, that do not systematically differentiate between spontaneous and indicated preterm birth, the latter being the most common delivery cause in the late preterm period” à Indicated preterm births were included in some studies?

• “Perinatal mortality and RDS were reduced among women treated with vaginal P , while data on 17-HP remain inconclusive” à contradicts the main results of your meta-analysis

• “Obviously, the higher is the gestational age at randomization, the lower is the possibility that a chronic treatment would decrease the event” à Does not make sense

• None of the studies have directly compared vaginal P and 17 HP à A network meta-analysis would have been better

Reviewer #2: The article is well written and clear on both the methodology and the results presented.

The limits of meta-analysis and their interest in clinical practice are well exposed in the article and by themselves show the limits of interpretation of the data contained in meta-analyses since it is an assembly of data coming from studies which themselves often contain methodological drawbacks.

Concerning the efficacy of 17OPH, I still find it very surprising that the 2 randomized trials that represent more than half of the meta-analysis (Rozenberg 2012, Faccinetti 2017), while methodologically well-conducted, conclude to a non efficacy of 17OPH in the reduction of prematurity and leads to a meta-analysis that concludes to an efficacy…. especially since the small randomized trials added are methodologically criticizable: open-label trial, small sample size, old data ...... Indeed, these 2 trials had the required power to show a difference on the primary outcome

The authors found no difference in preterm delivery before 34 weeks' gestation when comparing vaginal progesterone to placebo and no treatment, but did find a significant difference in this outcome when using 17OPH. On the other hand, they report a decrease in morbidity (decrease in low birth weight, neonatal respiratory distress syndrome) when using vaginal progesterone compared to placebo which is not the case when using 17OPH. How do the authors explain this when this morbidity is directly related to the complications of prematurity? These results are not consistent with each other. It is not reasonable to believe that vaginal progesterone improves morbidity by direct effect and not through improvement of prematurity. the authors have well explained the heterogeneity of doses, frequency of progesterone used in the different trials which makes it impossible to give clear instructions to clinicians, even with a positive result in favor of 17OPH .... the conclusion of the abstract should be that of the article which is that the data are still insufficient to change the medical practices of the use of 17OPH

Reviewer #3: This is a meta-analysis of randomized control trials examining prevention of preterm birth with vaginal and oral progestogens as well as 17-alpha hydroxyprogesterone caproate. The authors reported that they found a moderate certainty of evidence that 17-alpha hydroxyprogesterone caproate prevents preterm birth at less than 34 weeks gestation, and that they found a low certainty of evidence that oral progesterone prevents preterm birth at less than 37 weeks gestation.

The authors did not have enough information to analyze for potential confounding factors such as history of preterm birth. Additionally, differing dosages and study designs substantially limit this report. The inclusion of oral progesterone is less compelling and distracts from the findings of the report given the large amount of text presented. I would recommend removal of oral P as it is a distraction.

Additional comments to the report:

1. Line 74: The authors may consider defining maintenance tocolysis or describing the agents more typically used as tocolysis, and any potential theorized biologic basis for progesterone acting as a maintenance tocolysis

2. Line 75: which observational studies are the authors referring to?

3. How could you control for the different possible types of initial tocolytics used?

4. Was there a difference between studies that used the same progesterone but different initial tocolytics?

5. Line 256 – it is significant that only 1 study investigating 17 OHP reported history of preterm birth. This is a limitation of this analysis.

6. Line 281 – “our meta-analysis corroborated the suspect that vaginal P did not prevent PTB.” This is a strong conclusion to draw from data.

6. PLOS authors have the option to publish the peer review history of their article (what does this mean?). If published, this will include your full peer review and any attached files.

Reviewer #1: No

Reviewer #2: **Yes: **Senat MV

Reviewer #3: No

---

## [Author Response · Author response to Decision Letter 0]

21 Aug 2022

Response to Editor and Reviewers

Additional Editor Comments:

• The manuscript assess the role of maintenance therapy with two type of progesterone on preventing preterm birth less than 34 and 37 weeks of gestation. There are some major concerns raised by the reviewers and its mainly due to the inconsistency of the writing or the language used. How was the meta- analysis done when there are no direct comparison of these two types of therapy in any of the study included in the analysis?

Authors response:

the aim of our SR was not to directly compare the effect of the three types of progesterone but to “investigate the individual role played by 17-alpha hydroxyprogesterone caproate (17-HP), vaginal progesterone (Vaginal P) and oral progesterone (Oral P).” For this reason, we did not perform a Network meta-analysis, which would have allowed us to draw conclusion about the comparisons of these three types of progesterone versus each other, also in absence of direct comparison in the included trials. On the contrary, we performed pairwise meta-analyses comparing each of the different type of progesterone with placebo or no intervention. The methods to perform pairwise meta-analysis is described in the methods section- data extraction and analysis.

TEXT CHANGES

Abstract: line 25

“ As different progestogens have distinct molecular structures and biological effects, we conducted a systematic review and pairwise meta-analysis to investigate the individual role played by 17-alpha-hydroxyprogesterone caproate (17-HP), vaginal progesterone (Vaginal P) and oral progesterone (Oral P).”

Pag. 4, line 83

“Thus, the objective of this systematic review and pairwise meta-analysis is to evaluate the efficacy of maintenance tocolysis with 17-alpha-hydroxyprogesterone caproate (17-HP), vaginal progesterone (P) or oral P when compared to placebo or no treatment.”

5. Reviewers Comments to the Author

Reviewer #1:

• First of all how was preterm labor defined in these studies? How many patients were in established preterm labor with cervical changes and how many were only threatened preterm labor in each study?

Authors response:

We performed a detailed analysis of preterm labor definitions in primary studies (see attached table A) . Threatened preterm labour was homogenously defined as the simultaneous presence of regular contractions and cervical changes. Therefore, we expect that all the patients included show cervical changes at randomization.

TEXT CHANGES

Pag 5, line 104

“…treatment or placebo. Threatened preterm labour was homogenously defined, in the different studies, as the simultaneous presence of regular contractions and cervical changes.”

• Some of the studies have used vaginal P only up to 34 weeks, what is the point of studying delivery at 37 weeks in those studies?

Authors response:

Thank you for this comment. You are right! In two studies (Hyett 2020 and Stjernholm 2021) vaginal P prophylaxis was interrupted at 34 weeks. One of them, however, measured as primary outcome both PTB < 34 and PTB <37. Since vaginal P has a short half-life (<24 hrs) we cannot expect its efficacy to last longer. Therefore, we excluded Stjernholm et al 2021 from the calculation of the PTB <37, and related Forest Plot. Hyett 2020 did not reported PTB < 34 or PTB < 37. This study was therefore included only for some secondary outcome.

TEXT CHANGES

Pag 16, line 211

“Preterm birth < 37 weeks’ gestation did not differ among women receiving vaginal P as opposed to placebo/no treatment (RR 0.95, 95%CI 0.72, 1.26, 8 studies, 1231 participants , moderate certainty of evidence), while oral P significantly reduced the outcome when compared to placebo (RR 0.58, 95%CI 0.36 to 0.93, 1 study, 90 participants, low certainty of evidence).”

Fig. 4 is replaced with new forest plot , fig. 5

Tab S2 is replaced with new Tab S3 

• Subgroup analysis with regard to the initial drug used for arrest of preterm labor would be desirable.

Authors response:

Thank you for this comment. Trials are very heterogeneous for types of initial drug used as reported in Table 1. Most of the studies (13) used a single initial tocolytic drug. Five studies used Atosiban ( Areia 2013, Facchinetti 2007, Palacio 2016, Stjernholm 2021, Wood 2017), four studies used Magnesium sulphate (Borna 2008, Gargari 2012, Hyett 2020, Sharami 2010), two studies used Nifedipine (Choudhary 2014, Regmi 2012), one study used Ritodrine (Arikan 2011) and another one used Isoxysuprine Hydrochloride (Mishra 2014). However, 4 studies used more than one initial tocolytic agent. Briery 2014 used intravenous magnesium sulfate, oral calcium-channel blockers or antiprostaglandin; Facchinetti 2017 used atosiban, nifedipine or indometacin; Martinez de Tejada used B-mimetics, oxitocin receptor antagonist or calcium-channel blockers; finally Rozenberg used oral nifedipine, intravenous nicardipine or salbutamol.

Therefore, even if it would be very interesting it is not feasible to do a subgroup analysis with regard to the initial drug used for arresting preterm labor. Moreover, all those drugs were supposed to be effective for 48 hours, with no significant impact on maintenance tocolysis.

NO TEXT CHANGES

• “The apparent inconsistency may be due to a limitation of primary studies, that do not systematically differentiate between spontaneous and indicated preterm birth, the latter being the most common delivery cause in the late preterm period” à Indicated preterm births were included in some studies?

Authors response:

With “indicated preterm birth” we mean iatrogenic birth, defined as a birth that occurs before 37 weeks of gestation due to a planned delivery for maternal and/or adverse fetal conditions (induction of labor or cesarean section in the absence of spontaneous labor or rupture of membranes).

TEXT CHANGES

Pag 19, line 284

“The apparent inconsistency may be due to a limitation of primary studies, which do not systematically differentiate between spontaneous and iatrogenic PTB (defined as planned delivery for maternal and/or adverse fetal conditions), the latter being the most common delivery cause in the late preterm period37”.

• “Perinatal mortality and RDS were reduced among women treated with vaginal P, while data on 17-HP remain inconclusive” à contradicts the main results of your meta-analysis

Authors response:

Thank you for your comment. We amended the text, also according to the comments of Reviewer 2.

TEXT CHANGES

Pag. 19-20, line 304-310

Finally, it also seems that perinatal mortality and RDS were reduced among women treated with vaginal P, while data on 17-HP showed no difference 11. The apparent conflicting results between effects of the 2 progestogens on primary outcomes and lack of related effects on morbidity, could be ascribed to the small number of events included in the MA for low birth weight and RDS. The small number of events, together with the small sample size and the relatively small number of included studies strongly affect the precision of the estimate, returning in wide confidence intervals crossing the line of no effect.

• “Obviously, the higher is the gestational age at randomization, the lower is the possibility that a chronic treatment would decrease the event” à Does not make sense

Authors response:

The basic idea underlying this statement was that if gestational age at randomization is advanced, less is the time to allow the treatment effective for prophylaxis. Indeed, the time to primary outcome measure is reduced (2 weeks shorter). However, since such reasoning seems confusing, we delete the sentence from the manuscript.

TEXT CHANGES

Pag. 19, previous Lines 296-7: deleted

Moreover, to make more readable the comments on latency, the above corrected statement was insert.

TEXT CHANGES

Pag. 19-20, line 304-310

Finally, it also seems that perinatal mortality and RDS were reduced among women treated with vaginal P, while data on 17-HP showed no difference 11. The apparent conflicting results between effects of the 2 progestogens on primary outcomes and lack of related effects on morbidity, could be ascribed to the small number of events included in the MA for low birth weight and RDS. The small number of events, together with the small sample size and the relatively small number of included studies strongly affect the precision of the estimate, returning in wide confidence intervals crossing the line of no effect.

• None of the studies have directly compared vaginal P and 17 HP à A network meta-analysis would have been better.

Authors response:

As replied to the Editor additional comment, a Network meta-analysis would have allowed us to draw conclusion about the comparisons of the three types of progesterone versus each other also in absence of direct comparison in the included trials. However, as reported in manuscript, the aim of our SR was not to direct compare the effect of the three types of progestogens but to “investigate the individual role played by 17-alpha hydroxyprogesterone caproate (17-HP), vaginal progesterone (Vaginal P) and oral progesterone (Oral P).” This was further specified in the text.

TEXT CHANGES

Abstract: line 25

“ As different progestogens have distinct molecular structures and biological effects, we conducted a systematic review and pairwise meta-analysis to investigate the individual role played by 17-alpha-hydroxyprogesterone caproate (17-HP), vaginal progesterone (Vaginal P) and oral progesterone (Oral P).”

Pag. 4, line 83

“Thus, the objective of this systematic review and pairwise meta-analysis is to evaluate the efficacy of maintenance tocolysis with 17-alpha-hydroxyprogesterone caproate (17-HP), vaginal progesterone (P) or oral P when compared to placebo or no treatment.”

Reviewer #2: The article is well written and clear on both the methodology and the results presented.

The limits of meta-analysis and their interest in clinical practice are well exposed in the article and by themselves show the limits of interpretation of the data contained in meta-analyses since it is an assembly of data coming from studies which themselves often contain methodological drawbacks.

• Concerning the efficacy of 17OPH, I still find it very surprising that the 2 randomized trials that represent more than half of the meta-analysis (Rozenberg 2012, Faccinetti 2017), while methodologically well-conducted, conclude to a non efficacy of 17OPH in the reduction of prematurity and leads to a meta-analysis that concludes to an efficacy…. especially since the small randomized trials added are methodologically criticizable: open-label trial, small sample size, old data ...... Indeed, these 2 trials had the required power to show a difference on the primary outcome

Authors response:

Thank you for your comments. We assessed the risk of bias of the included studies by the Cochrane risk of bias tool. We included the risk of bias assessment in the evaluation of the certainty of evidence of our results using the GRADE methodology, therefore we accounted for the limitation of the data included in meta-analyses. However, according to our evaluation (see risk of bias table in fig 2), Rozenberg 2012, Facchinetti 2017 were not substantially different in risk of bias to the other studies; we did not plan to exclude studies at high risk of bias from MA or to perform sensitivity analysis without high-risk studies and in any case we didn’t find any serious limitation in some studies that could have justified their exclusion because of risk of bias. Rozenberg 2012, Facchinetti 2017 are open label as several other studies, and we did not think that lack of blinding could seriously bias the objective outcomes evaluated in this review. Finally, the small sample size is not an issue when the data of different studies are pooled in MA, as the small studies have less weight in the MA.

NO TEXT CHANGES

• The authors found no difference in preterm delivery before 34 weeks' gestation when comparing vaginal progesterone to placebo and no treatment, but did find a significant difference in this outcome when using 17OPH. On the other hand, they report a decrease in morbidity (decrease in low birth weight, neonatal respiratory distress syndrome) when using vaginal progesterone compared to placebo which is not the case when using 17OPH. How do the authors explain this when this morbidity is directly related to the complications of prematurity? These results are not consistent with each other. It is not reasonable to believe that vaginal progesterone improves morbidity by direct effect and not through improvement of prematurity.

Authors response:

Thank you, we agree. The apparent conflicting results between favorable effect of 17-HP on preterm delivery before 34 weeks and no effect on morbidity could be ascribed to the small number of events included in the MA for low birth weight and respiratory distress syndrome. We add a sentence on this topic.

Moreover, in the discussion section we already acknowledge limitations declaring that we could draw definite conclusions only on the primary outcomes (line 318).

TEXT CHANGES

Pag. 19-20, line 304-310

Finally, it also seems that perinatal mortality and RDS were reduced among women treated with vaginal P, while data on 17-HP showed no difference 11. The apparent conflicting results between effects of the 2 progestogens on primary outcomes and lack of related effects on morbidity, could be ascribed to the small number of events included in the MA for low birth weight and RDS. The small number of events, together with the small sample size and the relatively small number of included studies strongly affect the precision of the estimate, returning in wide confidence intervals crossing the line of no effect.

• the authors have well explained the heterogeneity of doses, frequency of progesterone used in the different trials which makes it impossible to give clear instructions to clinicians, even with a positive result in favor of 17OPH .... the conclusion of the abstract should be that of the article which is that the data are still insufficient to change the medical practices of the use of 17OPH

Authors response:

Thank you for this comment, you are right, although from our study the 17-HP appears to prevent PTB we haven’t strong evidence to indicate its use in clinical practice. Indeed, in Discussion Section (line 328) we wrote: “However there are insufficient data to generate recommendations for the use of progestagens as a maintenance tocolytic in clinical practice”. This conclusion will be added to the abstract.

TEXT CHANGES

Abstract: line 47-48

“With a moderate certainty of evidence, 17-HP prevents PTB <34 weeks’ gestation among women that remained undelivered after an episode of threatened preterm labour. However, data are insufficient to generate recommendations in clinical practice. In the same women, both 17-HP and vaginal P are ineffective in the prevention on PTB <37 weeks.”

Reviewer #3: This is a meta-analysis of randomized control trials examining prevention of preterm birth with vaginal and oral progestogens as well as 17-alpha hydroxyprogesterone caproate. The authors reported that they found a moderate certainty of evidence that 17-alpha hydroxyprogesterone caproate prevents preterm birth at less than 34 weeks’ gestation, and that they found a low certainty of evidence that oral progesterone prevents preterm birth at less than 37 weeks gestation.

The authors did not have enough information to analyze for potential confounding factors such as history of preterm birth. Additionally, differing dosages and study designs substantially limit this report.

• The inclusion of oral progesterone is less compelling and distracts from the findings of the report given the large amount of text presented. I would recommend removal of oral P as it is a distraction.

Authors response:

The aim of our systematic review was to assess the efficacy of the various existing types and route of administration of progesterone, through a comprehensive systematic approach searching for all existing evidence. According to this systematic approach is not appropriate to exclude ex post one drug for the only reason that only one study has been retrieved for this type of intervention.

NO TEXT CHANGES

Additional comments to the report:

• Line 74: The authors may consider defining maintenance tocolysis or describing the agents more typically used as tocolysis, and any potential theorized biologic basis for progesterone acting as a maintenance tocolysis

Authors response:

We thank the reviewer. We changed the text as follow:

TEXT CHANGES

Pag. 4, line 67

“Acute tocolysis over the first 48 hours is an established practice to delay delivery until antenatal corticosteroids are administered and/or pregnant women are transferred to a tertiary care center.”

Pag 4, line 69

“The efficacy of tocolysis beyond the 48 hours (also known as maintenance or secondary tocolysis) is still debated3. Indeed, many of the agents tested (magnesium, beta-agonists, etc..) provided no efficacy4.”

Finally, as long as any potential biologic basis for progesterone acting as a maintenance tocolytic we add:

TEXT CHANGES

Pag 4, line 76

“Several plausible mechanisms of progesterone supplementation to prevent PTB have been proposed, including reduced gap-junction formation, oxytocin antagonism (leading to uterine relaxation), maintenance of cervical integrity and anti-inflammatory effects8.”

• Line 75: which observational studies are the authors referring to?

Authors response:

Thank you for this comment, you are right, in line 75 we wrongly mentioned “observational studies”, instead of small randomized controlled trials. Therefore, we amend in the text.

TEXT CHANGES

Pag 4, line 79

“Progestogens for maintenance tocolysis were shown to effectively delay delivery according to small low quality randomized controlled trials, but not in larger high quality RCTs9.”

• How could you control for the different possible types of initial tocolytics used?

Authors response:

To control for the different possible types of initial tocolytics a subgroup analysis would be required. We specifically replied to reviewer #1. Briefly, the heterogeneity for initial tocolytics used (more than one drug in some studies) makes it not feasible doing the subgroup analysis with regards to the initial drug used for arrest of preterm labor. Moreover, all these drugs for acute tocolysis were supposed to be efficacious for 48 hours, with no significant impact on maintenance tocolysis.

NO TEXT CHANGES

• Was there a difference between studies that used the same progesterone but different initial tocolytics?

Authors response:

Trials are heterogeneous for initial tocolytics used (see Table 1). Therefore, even if it would be very interesting and useful it is not feasible to check the difference between studies that used the same progesterone but different initial tocolytics

NO TEXT CHANGES

• Line 256 – it is significant that only 1 study investigating 17 OHP reported history of preterm birth. This is a limitation of this analysis.

Authors response:

The history of preterm birth is one of the most important risk factor for PTB and is reported only in 1 study investigating 17-HP. We already acknowledge this limitation of meta-analysis and it is the reason why we could not perform the subgroup analysis on the history of PTB on women treated with 17-HP, (mentioned in line 318 of the discussion section).

NO TEXT CHANGES

• Line 281 – “our meta-analysis corroborated the suspect that vaginal P did not prevent PTB.” This is a strong conclusion to draw from data.

We better specified.

TEXT CHANGES

Pag 19, line 287

“Based on a large study population (>2000 women) deriving from different countries, our meta-analysis also corroborated the previous suspect that vaginal P has not helpful for PTB prevention after an episode of preterm labour 9,39,40”.

---

## [Decision Letter · Decision Letter 1]

27 Sep 2022

PONE-D-22-09216R1PROGESTOGENS FOR MAINTENANCE TOCOLYSIS IN SYMPTOMATIC WOMEN. A SYSTEMATIC REVIEW AND META-ANALYSISPLOS ONE

Dear Dr. Ferrari,

Thank you for submitting your manuscript to PLOS ONE. After careful consideration, we feel that it has merit but does not fully meet PLOS ONE’s publication criteria as it currently stands. Therefore, we invite you to submit a revised version of the manuscript that addresses the points raised during the review process.

ACADEMIC EDITOR:thank you for the revision Authors need to address the issues raised by the reviewer as it affect the scientific interpretation of the results

We look forward to receiving your revised manuscript.

Kind regards,

Anish Keepanasseril

Academic Editor

PLOS ONE

Reviewers' comments:

Reviewer's Responses to Questions

**Comments to the Author**

1. If the authors have adequately addressed your comments raised in a previous round of review and you feel that this manuscript is now acceptable for publication, you may indicate that here to bypass the “Comments to the Author” section, enter your conflict of interest statement in the “Confidential to Editor” section, and submit your "Accept" recommendation.

Reviewer #3: (No Response)

2. Is the manuscript technically sound, and do the data support the conclusions?

Reviewer #3: Partly

3. Has the statistical analysis been performed appropriately and rigorously? 

Reviewer #3: I Don't Know

4. Have the authors made all data underlying the findings in their manuscript fully available?

Reviewer #3: Yes

5. Is the manuscript presented in an intelligible fashion and written in standard English?

Reviewer #3: Yes

6. Review Comments to the Author

Reviewer #3: Thank you for the thorough comments and response to the reviewers queries. There remains a critical element needed for clarification.

Can the authors please further clarify the comments to Reviewer #2, Concerning the efficacy of 17OPH, I still find it very surprising that the 2 randomized trials that represent more than half of the meta-analysis (Rozenberg 2012, Faccinetti 2017), while methodologically well-conducted, conclude to a non efficacy of 17OPH in the reduction of prematurity and leads to a meta-analysis that concludes to an efficacy…. especially since the small randomized trials added are methodologically criticizable: open-label trial, small sample size, old data ...... Indeed, these 2 trials had the required power to show a difference on the primary outcome.

The authors state in Authors response:

Thank you for your comments. We assessed the risk of bias of the included studies by the Cochrane risk of bias tool. We included the risk of bias assessment in the evaluation of the certainty of evidence of our results using the GRADE methodology, therefore we accounted for the limitation of the data included in meta-analyses. However, according to our evaluation (see risk of bias table in fig 2), Rozenberg 2012, Facchinetti 2017 were not substantially different in risk of bias to the other studies; we did not plan to exclude studies at high risk of bias from MA or to perform sensitivity analysis without high-risk studies and in any case we didn’t find any serious limitation in some studies that could have justified their exclusion because of risk of bias. Rozenberg 2012, Facchinetti 2017 are open label as several other studies, and we did not think that lack of blinding could seriously bias the objective outcomes evaluated in this review. Finally, the small sample size is not an issue when the data of different studies are pooled in MA, as the small studies have less weight in the MA.

NO TEXT CHANGES

Specifically, how can the Briery 2014 report have "not substantially different risk of bias" when the enrollment of the entire study did not reach a priori sample size requirements because the fellow graduated when compared to the other studies? In the Briery study, 45 singleton pregnancies were randomized after successful tocolysis; however, the methods stated, "sample size calculation suggested that 80 treated pregnancies (40 in each group) would yield a sufficient number to have an 80% power of detecting a 20% reduction in delivery before 37 weeks." That is, this study did not enroll a prior sample size needed for a primary outcome of <37 weeks.

The small sample is less weighted but carries the bulk of "benefit" of 17OHP-C on a secondary outcome of an underpowered study. This is a key element in the analysis as this drives the final conclusions for 17OHP-C.

7. PLOS authors have the option to publish the peer review history of their article (what does this mean?). If published, this will include your full peer review and any attached files.

Reviewer #3: No

---

## [Author Response · Author response to Decision Letter 1]

21 Oct 2022

We thank the Reviewer for the comment.

We reported that studies, including Briery 2014, have "not substantially different risk of bias" because the sample size and the ability or not of the researchers to achieve the predetermined sample size to have an adequate power does not introduce bias, according to the Cochrane Criteria, the most accepted and universally used tool to assess RoB of studies included in systematic reviews. As a matter of fact, Briery 2014 study was included in all the previous metanalyses ( 9; 11; 40). The small sample size however impacts on the precision of the estimate of this study, not on risk of bias. Therefore, we accept your comment and added to the text that the precision of the estimate could be reduced, due to the small number of events (see text changes in the discussion section).

Finally, within the perspective of a SR and meta-analysis, it does not matter whether the primary studies defined an outcome as primary or secondary. All the data are pooled and studies possibly underpowered to detect a difference because the outcome was defined as secondary, will contribute less to the overall estimate.

Text changes:

Line 276: “It should be underlined that the relatively small number of events impacts on the precision of the estimate.”

---

## [Decision Letter · Decision Letter 2]

31 Oct 2022

PROGESTOGENS FOR MAINTENANCE TOCOLYSIS IN SYMPTOMATIC WOMEN. A SYSTEMATIC REVIEW AND META-ANALYSIS

PONE-D-22-09216R2

Dear Dr. Ferrari,

We’re pleased to inform you that your manuscript has been judged scientifically suitable for publication and will be formally accepted for publication once it meets all outstanding technical requirements.

Kind regards,

Huijuan Cao, Ph.D.

Academic Editor

PLOS ONE

Additional Editor Comments (optional):

Reviewers' comments:

Reviewer's Responses to Questions

**Comments to the Author**

1. If the authors have adequately addressed your comments raised in a previous round of review and you feel that this manuscript is now acceptable for publication, you may indicate that here to bypass the “Comments to the Author” section, enter your conflict of interest statement in the “Confidential to Editor” section, and submit your "Accept" recommendation.

Reviewer #3: All comments have been addressed

2. Is the manuscript technically sound, and do the data support the conclusions?

Reviewer #3: Yes

3. Has the statistical analysis been performed appropriately and rigorously? 

Reviewer #3: Yes

4. Have the authors made all data underlying the findings in their manuscript fully available?

Reviewer #3: Yes

5. Is the manuscript presented in an intelligible fashion and written in standard English?

Reviewer #3: Yes

6. Review Comments to the Author

Reviewer #3: Thank you for reviewing and addressing concerns. Given the recent US FDA events on 17OHP-C, this report is of importance and interest.

7. PLOS authors have the option to publish the peer review history of their article (what does this mean?). If published, this will include your full peer review and any attached files.

Reviewer #3: No

---

## [Editor Report · Acceptance letter]

7 Nov 2022

PONE-D-22-09216R2 

Progestogens for maintenance tocolysis in symptomatic women. A systematic review and meta-analysis 

Dear Dr. Ferrari:

I'm pleased to inform you that your manuscript has been deemed suitable for publication in PLOS ONE. Congratulations! Your manuscript is now with our production department. 

Kind regards, 

on behalf of

Dr. Huijuan Cao 

Academic Editor

PLOS ONE